# The effectiveness and characteristics of mHealth interventions to increase adolescent's use of Sexual and Reproductive Health services in Sub-Saharan Africa: A systematic review

Franklin I. Onukwugha[1]*, Lesley Smith[1], Dan Kaseje[2], Charles Wafula[2], Margaret Kaseje[2], Bev Orton[3], Mark Hayter[4], Monica Magadi[3]

**1** Institute for Clinical and Applied Health Research (ICAHR), Faculty of Health Sciences, University of Hull, Hull, United Kingdom, **2** Tropical Institute of Community Health and Development (TICH), Kisumu, Kenya, **3** Faculty of Arts, Culture and Education, University of Hull, Hull, United Kingdom, **4** Faculty of Health, Psychology & Social Care, Manchester Metropolitan University, Manchester, United Kingdom

* f.i.Onukwugha@hull.ac.uk

## Abstract

### Background

mHealth innovations have been proposed as an effective solution to improving adolescent access to and use of Sexual and Reproductive Health (SRH) services; particularly in regions with deeply entrenched traditional social norms. However, research demonstrating the effectiveness and theoretical basis of the interventions is lacking.

### Aim

Our aim was to describe mHealth intervention components, assesses their effectiveness, acceptability, and cost in improving adolescent's uptake of SRH services in Sub-Saharan Africa (SSA).

### Methods

This paper is based on a systematic review. Twenty bibliographic databases and repositories including MEDLINE, EMBASE, and CINAHL, were searched using pre-defined search terms. Of the 10, 990 records screened, only 10 studies met the inclusion criteria. The mERA checklist was used to critically assess the transparency and completeness in reporting of mHealth intervention studies. The behaviour change components of mHealth interventions were coded using the taxonomy of Behaviour Change Techniques (BCTs). The protocol was registered in the 'International Prospective Register for Systematic Reviews' (PROSPERO-CRD42020179051).

**Data Availability Statement:** All relevant data are within the manuscript and its Supporting Information files.

**Funding:** QR-GCRF pump-priming fund from the University of Hull, UK. EAL012. The funders had no role in study design, data collection and analysis, decision to publish, or preparation of the manuscript.

**Competing interests:** The authors have declared that no competing interests exist.

## Results

The results showed that mHealth interventions were effective and improved adolescent's uptake of SRH services across a wide range of services. The evidence was strongest for contraceptive use. Interventions with two-way interactive functions and more behaviour change techniques embedded in the interventions improved adolescent uptake of SRH services to greater extent. Findings suggest that mHealth interventions promoting prevention or treatment adherence for HIV for individuals at risk of or living with HIV are acceptable to adolescents, and are feasible to deliver in SSA. Limited data from two studies reported interventions were inexpensive, however, none of the studies evaluated cost-effectiveness.

## Conclusion

There is a need to develop mHealth interventions tailored for adolescents which are theoretically informed and incorporate effective behaviour change techniques. Such interventions, if low cost, have the potential to be a cost-effective means to improve the sexual and reproductive health outcomes in SSA.

## Background

Globally, adolescents and young people face enormous barriers accessing Sexual and Reproductive Health (SRH) information and services [1–3], especially in Sub-Saharan Africa (SSA) with a high burden of HIV/AIDS and unintended pregnancy [4–6]. These barriers such as lack of awareness of available services, lack of confidentiality, service providers attitude, social norms and values and restrictive policies, operate at different levels [5, 6].

Adolescent girls and young women accounted for 25% of all new HIV infections globally in 2017 and of all HIV infections occurring among adolescents in SSA; 80% are in girls aged 15–19 years [7]. Sub-Saharan Africa had the highest prevalence of adolescent pregnancy in the world, between 1995 and 2011, with an estimated 104 births per 1,000 women aged 15–19 [8, 9]; and young women aged 15–24 years account for 57% of abortions [10, 11].

With over 600 million mobile phone subscribers predicted by 2025 in SSA [12], representing about half of the population, mHealth innovations have been proposed as a solution to improving access to and use of health services among the underserved population, especially in settings with poor healthcare infrastructure [13, 14]. Mobile health or mHealth is defined as a medical and public health practice supported by mobile phones, tablets, patient monitoring devices, personal digital assistants (PDAs), and other wireless devices [14]. mHealth can offer timely, accurate and non-judgemental SRH information and services to adolescents [15]. A systematic review identified 487 mHealth programmes implemented in SSA between 2006 and 2016 [16], although few involved adolescent SRH. Furthermore, most programmes in the region have not been rigorously evaluated [17].

Previous reviews have collated and evaluated mHealth interventions to improve adolescent's uptake of SRH services [18–20]. The reference to SRH in this paper is consistent with the United Nations definition: a state of complete physical, mental and social well-being in all matters relating to reproduction, enabling people to have a satisfying safe sex life and the freedom to decide if, when and how often to reproduce, which implies the right of men and women to be informed and to have access to safe, effective, affordable and acceptable methods of family planning of their choice [21]. While countries have expanded their vision of

addressing people's rights to a full and comprehensive range of SRH services (including reproductive cancers, gender-based violence, etc), we recognize that many developing countries, especially sub-Saharan Africa, are only able to offer a core package of basic SRH services [22]. Therefore, we focus on the core / basic SRH, including family planning/contraception, sexually transmitted infections (including HIV/AIDS), and pregnancy/termination-related issues.

Smith and colleagues provided some evidence that interventions delivered by mobile phones improve contraception use, although none of the studies were carried out in SSA [18]. Evidence from a systematic review study showed that health promotion campaigns implemented with text messaging improved SRH knowledge, reduced unprotected sex, and increased STI testing among adolescents [19]. However, only three out of 35 studies in the review were based in low- and middle-income countries (LMICs). A more recent review which used SRH defined as access to comprehensive sexuality education; services to prevent, diagnose and treat STIs and counselling on family planning, and found mHealth interventions to be effective in improving uptake of antenatal care and postnatal care services, especially those that were aimed at changing behaviour of pregnant women [20]. However, the study reported paucity of evidence on other types of mHealth applications.

Although these reviews shed some light on the quality of mHealth evidence, approaches, and barriers, in improving adolescents' access to SRH services in LMICs, there is a dearth of understanding on the effectiveness of mHealth interventions in improving uptake of services specifically among adolescents in SSA. Our review builds on previous evidence by exploring the theoretical and empirical basis of mHealth interventions using a taxonomy of behaviour change techniques [23], and assesses the completeness of reporting mHealth interventions for improving adolescent's uptake of SRH services using the WHO developed mERA checklist [24].

### Research aim

To determine the effectiveness of mHealth interventions to improve the uptake of sexual and reproductive health (SRH) services by adolescents in Sub-Saharan Africa.

### Primary objectives are to

i. Describe the components of mHealth interventions addressing SRH among adolescents in SSA

ii. Assess the effectiveness of mHealth interventions addressing SRH among adolescents in SSA

### Secondary objectives are to assess the

i. Acceptability of mHealth interventions to adolescents, and parents in providing SRH information in SSA.

i. Feasibility of delivery of mHealth interventions by providers

ii. Cost-effectiveness of mHealth interventions in SSA.

### Methods

This review was carried out using the guidance developed by the Centre for Reviews and Dissemination [25]. The protocol was registered in the 'International Prospective Register for Systematic Reviews' (PROSPERO) CRD42020179051 [26]

The review was based on evaluation studies that assessed the effectiveness of mHealth interventions to support the delivery of information, decision-making, behaviour change or risk reduction strategies regarding SRH among adolescents aged 10–19 years. Although, our target population was adolescents aged 10–19 years [6], however, interventions that focus on young people aged 10–24 years was considered also as interventions that focus on young people aged 10–24 may likely not be different from those of adolescents aged 10–19 years.

## Inclusion and exclusion criteria

This review was based on studies conducted in Sub-Saharan Africa (SSA) as defined by the UN Development Program 2020 [27]. We included any single or multi-component mHealth/ mobile health interventions that supports delivery of information, decision-making, behaviour change or risk reduction strategies regarding Sexual and Reproductive Health (SRH). We included evaluation studies such as Randomized control trials (RCTs), other experimental and quasi/non-experimental studies that assess the effectiveness of mHealth interventions. Studies outside these parameters were not considered.

## Electronic searches

Eight primary bibliographic databases (MEDLINE, EMBASE, CINAHL, PsycINFO, Web of Science, Cochrane Library Central Register of Controlled Trials, SCOPUS and Academic search premier), six institutional digital databases (WHO Global Health Library, African population and health research centre (APHRC), United Nations Population Fund (UNFPA), Guttmacher Institute, Population Council, and Family Health International) and other repositories (ProQuest, International Bibliography of social sciences, OpenDOAR, Ethos-British Library, Network digital library of Thesis and Dissertation and ZETOC) were searched from April to May 2020 for peer-reviewed articles and grey literature. There was no restriction in terms of language or publication year. Non-English language published papers on mHealth were not identified during our literature search.

## Search strategy

The search strategy was developed by FO with input from LS, DK and MM. Search terms were iteratively developed within each of three search concepts: (i) Sexual and Reproductive Health; (ii) mHealth; (iii) Sub-Saharan Africa. The keywords and database thesaurus terms were combined one after the other using Boolean Operators and truncation/wildcards were applied and modified where appropriate. Full details of the review protocol is published online [26] and the full search strategy available as S1 File.

Study records were exported to Endnote and titles and abstracts screened by FO and MK and disagreement resolved by MM. Full-text articles were independently screened for inclusion by three reviewers (FO, CW and MH) using Rayyan QCRI software and disagreement resolved by discussion with LS.

## Data extraction

Data extracted were the author's name and year, study design & sample size, study settings, interventions, target population and outcomes. Table 1 provides the full sample description.

Table 2 & Table 3 summarise the results of each paper. The WHO developed mHealth Evidence Reporting and Assessment (mERA) Checklist comprising 16 items focused on reporting mHealth interventions was used to critically assess the content, context, implementation features and completeness in reporting of mHealth studies [24]. Also, the behavioural change

**Table 1. Sample description.**

| Authors & year/ country | Design | Sample size | Settings | Target population | Interventions | control | Outcomes |
|---|---|---|---|---|---|---|---|
| De-Kruijf et al 2016<br><br>Ghana<br><br>Dutch (Netherlands) | Survey & Focus Group Discussion | 172 | Community-based | Males & females<br><br>14–23 years | SMS in community where peer educators have phones | Community where peer educators have no mobile phone. | SRH knowledge about STIs, abortion & contraception |
| Ivanova 2019<br><br>Kenya | Pre-post design | 90 | Hospital +Clinic) | Males & females<br><br>15–24 years, | An interactive web-based intervention with posts and discussions. | Non web-based intervention | SRH knowledge, adherence intentions and feasibility/ acceptability |
| Rokicki et al., 2016, Rokicki and Fink 2017<br><br>Ghana | RCT | 756 | School-based | Females aged 14–24 years & among those in senior high schools | Interactive weekly text SMS on SRH | Uni-directional weekly SMS on Malaria. | SRH knowledge, Self-reported pregnancy, sexual activity, and contraceptive use |
| Harrington et al., 2019<br><br>Kenya | RCT | 260 | Hospitals | Females 14 years and above | SMS sent weekly on SRH | No SMS (Received standard care) | Contraceptive use, Exclusive breastfeeding, FP satisfaction, Contraceptive discontinuation and time to first initiation of any method. |
| L'Engle et al., 2013<br><br>Tanzania | Evaluation (Quasi experiment) | 506 | Population based (General public) | Males and females<br><br>< 19–40 years, | Interactive and menu-based SMS system. | No SMS intervention | Feasibility, contraceptive |
| Unger et al., 2018<br><br>Kenya | RCT | 300 | Clinics | Females,<br><br>14 years or older, | SMS weekly motivational message on maternal health | No SMS-Received standard care | Facility delivery; Exclusive Breastfeeding (EBF) and contraceptive use, maternal/ infant mortality |
| Linnemayr et al., 2017<br><br>Uganda | RCT | 332 | HIV hospitals | Males & females<br><br>Aged 15 to 22 years, | Bidirectional weekly messages on ART | Standard care (control) | Adherence to ART |
| MacCarthy et al., 2020<br><br>Uganda | Mixed (RCT + Qualitative) | 179 | HIV clinics | Males/females<br><br>15–24 years, | Own adherence+ peer adherence information | No intervention. | ART adherence, feasibility, acceptability. |
| Pintye et al., 2020<br><br>Kenya | Mixed (Questionnaire +client health information form) | 334 | Family planning Clinics | (HIV positive/ pregnant/ postpartum women Aged 18–30+ | Bidirectional weekly SMS in adherence encouragement, & self-efficacy | Standard care (No SMS) | PrEP Continuation/ Adherence/acceptability |
| Cele and Archary 2019<br><br>South Africa | Cross sectional | 100 | Hospital | Males and females<br><br>aged 12–19 years | SMS ART adherence Urban areas | ART adherence in rural areas. | Acceptability and feasibility |

components of the mHealth interventions were coded using the taxonomy of Behavioural Change Techniques (BCTs) [23]. Data extraction was completed by FO and checked by LS.

## Risk of bias (quality) assessment

Studies were appraised for methodological rigour/quality using the revised Cochrane risk-of-bias tool for randomized trials and the ROBINS-I tool for non-randomized studies [28]. Risk of bias assessment was completed by FO and independently reviewed by LS. RCTs were assessed based on random sequence generation (selection bias), allocation concealment, blinding, incomplete data, selective reporting, and other biases encountered throughout the study. The potential sources of bias for all the non-randomized studies were assessed based on the seven domains (selection of participants, measurement of interventions, departures from

**Table 2. Summary of results of included studies on mHealth interventions on SRH knowledge, sexual behaviour & contraceptive use.**

| Author's names & year | Method of assessment | Time points assessed | SRH knowledge | Sexual behaviour | Contraception/birth control |
|---|---|---|---|---|---|
| De Kruift 2016 | Survey (free recall of text messages, cued recall, and a knowledge test). | 3-months | No evidence of increase in Knowledge. The control group provided more correct answers about Sexually Transmitted Infections (STIs), abortion and contraception than the intervention group. | NA | NA |
| Ivanova 2019 | Using 17 true/false items adapted questions | 3-months | Improved SRH knowledge by 0.3 points. But was statistically significant for two items only (Wilcoxon signed ranks test–0.26). | NA | NA |
| Rokicki et al., 2016; Rokicki and Fink 2017 | Self-administered questionnaire | 3 & 15 months | Both unidirectional and interactive interventions increased knowledge at 3 & 15 months than in the control group. But knowledge level was higher in the interactive group. | The interactive intervention increased risk of sex without a condom (OR = 3.47; 95% CI = 1.12, 10.74). There was no impact on age of sexual debut for those who have ever had sexual intercourse. | Interactive intervention increased the use of a birth control pill (OR = 13.23; 95% CI = 1.08, 161.80) and decreased the likelihood of using emergency contraception. |
| Harrington et al., 2019 | Self-administered questionnaire | 6 weeks, 14 weeks, and 6-months | NA | Most women resumed sexual intercourse by 6 months (31.8% at 6 weeks, 57.9% at 14 weeks, and 67.7% at 6 months). | At 6-months, family planning initiation was higher in the intervention group but not sig. (0.74 vs 0.65; P = .12). Similar at 6- & 14-weeks post-partum. |
| Unger et al., 2018 | Self-reported questionnaire | 10 weeks, 16 weeks, and 6 months | NA | NA | At 16 weeks postpartum, contraceptive use was significantly higher in both intervention groups than in the control but statistically not significant at 6 months. |
| L'Engle et al., 2013 | Through electronic and automatic open-ended questions. | 10-months | NA | NA | Contraceptive use was higher among participants who engaged with the intervention than those who did not engage with the intervention (Engagement = 2.3; & non engagement = 1.4). |

intended interventions, the control of cofounders, missing data, and selection of reported results) of the ROBINs-I tool. More details of the risk of bias for the studies are provided under the section 'Characteristics of studies'.

## Strategy for data synthesis

The results of the search were reported and presented in a Preferred Reporting Items for Systematic Reviews and Meta-analyses (PRISMA) flow diagram. The extracted data for each included study was presented in tabular form, and results of the individual studies were narratively synthesised aligning with the review objectives as statistical pooling was not carried out due to the variation in study designs, interventions, sample population and outcome measures. See the review protocol for information on the strategy for data synthesis (CRD42020179051) [26].

## Results

### Study identification and selection

The search identified 10,990 citations. After removing duplicates, 6,401 citations were included for title and abstract screening of which 86 full-text articles were assessed yielding 10 studies

**Table 3. Summary of results of included studies on mHealth interventions on ART, pregnancy & childbirth and breast feeding.**

| Author's names & year | Method of assessment | Time points assessed | HIV treatment/ART adherence | Pregnancy & childbirth | Breast feeding |
|---|---|---|---|---|---|
| Linneneyr et al., 2017 | Electronic medication event monitoring system (MEMS) cap. | 48 weeks/ 12months | At 12-months, the adherence at 90% showed no statistical difference between the intervention groups compared with the control. Mean adherence was 64% for the 1-way group compared with 67% in the control group. | NA | NA |
| Pintye 2020 | Self-administered questionnaires | 10-months | Women who enrolled in the intervention (mWACh), reported higher pre-exposure prophylaxis (PrEP) adherence (73%) compared to 55% of women who initiated PrEP before the intervention (P < .001). The results remained significant after controlling for age and marital status (P = .003). | NA | NA |
| Ivanova 2019 | Using 17 true/false items adapted questions | 3-months | Post intervention participants reported higher (77.8%) adherence of ART than at baseline (71.6%). However, this was not statistically significant (p = 0.95). | NA | NA |
| MacCarthy 2020 | Surveys were used to record beliefs/ behaviours related. | 9-months | Adherence was 81.1% in the control group, 76.5% in intervention-T1 group, and 82.5% in the intervention-T2 group. After controlling for baseline adherence, the T1 group had 3.8%-point lower adherence than the control group (95% CI -9.9, 2.3) and the T2 group had 2.4% points higher adherence than the control group (95% CI -3.0, 7.9). | NA | NA |
| Harrington et al., 2019 | Self-administered questionnaire | 6 weeks, 14 weeks, and 6-months | NA | At 6-months, fertility intentions were similar between groups. 26.2% reported a desire to stop childbearing, and among 184 who wanted to become pregnant again, 88.6%) preferred to delay the next pregnancy by at least 3 years. | Exclusive breastfeeding was similar between groups at all the time points. |
| Unger et al., 2018 | Self-reported questionnaire | 10 weeks, 16 weeks, and 6 months | NA | At both 10wks and 16wks, facility delivery was not statistically significant across the 3 arms. Also, there were fewer stillbirths and infant deaths in the 2-way group compared to the control group but not statistically significant. | One-way and 2-way SMS groups at 10, 16 and 24 weeks improved EBF practices. However, the differences were not statistically significant. |

that met the review inclusion criteria. One RCT was published in two separate papers [29, 30], however, the main trial findings were reported in one [29] which was used throughout the review. See Fig 1 (PRISMA flow diagram).

## Study setting and design

Four studies were carried out in Kenya [31–34], two in Ghana [29, 35], one in South Africa [36], two in Uganda [37, 38], and one in Tanzania [39]. Five studies were RCTs [29, 33, 34, 37, 38], three were evaluation and pre-post design studies [31, 32, 39], and two were mixed

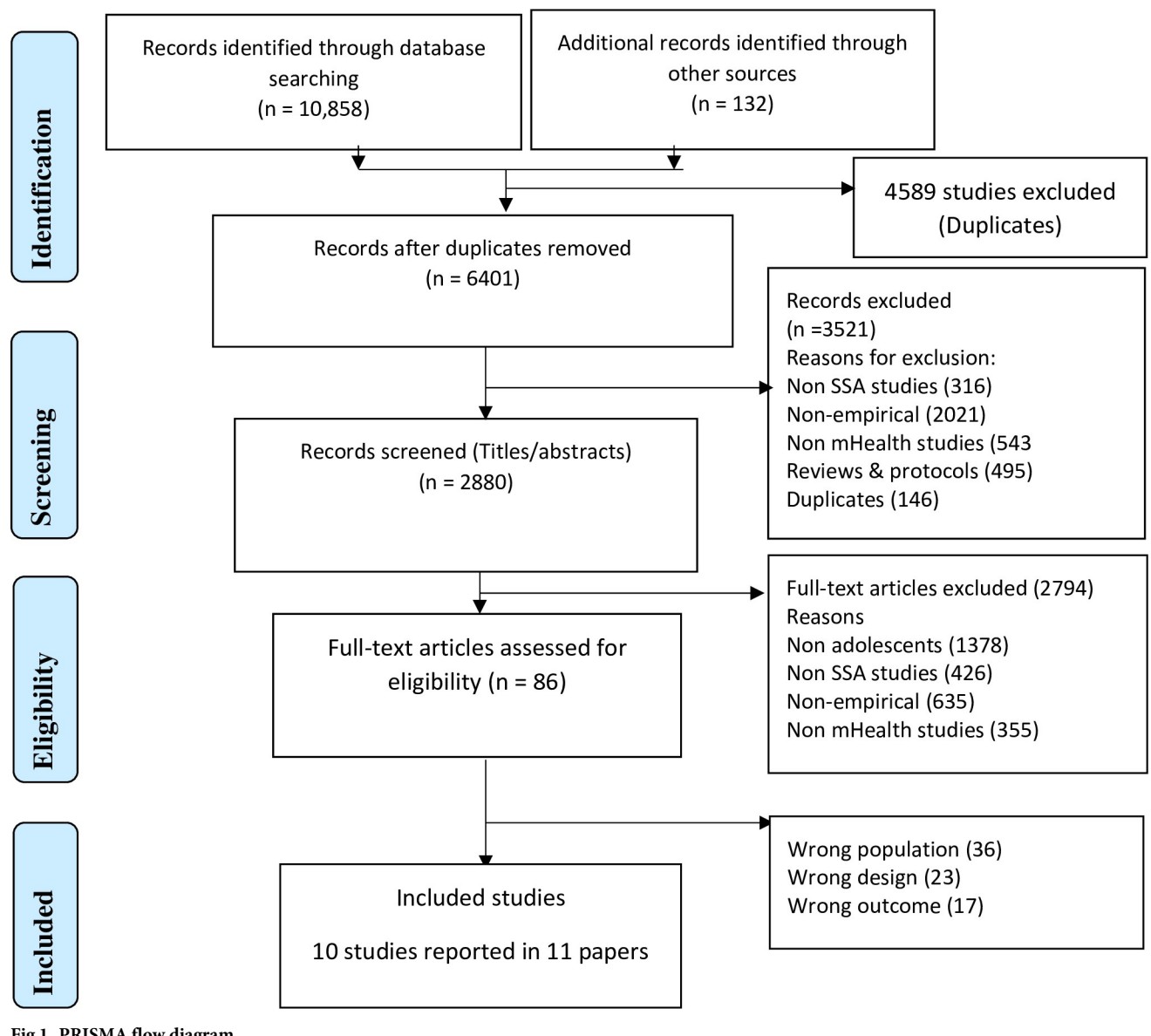

**Fig 1. PRISMA flow diagram.**

methods/cross-sectional studies [35, 36]. Seven studies occurred in a hospital setting [31–32, 36–38], one was school-based [29] and two were community-based [35, 39] (Table 1).

## Characteristics of studies

All 5 RCTs were judged at low risk of bias for most criteria [29, 33, 34, 37, 38]. All reported random sequence generation except for one study [38], and all reported adequate method of concealment, losses to follow up and prespecified outcomes [29, 33, 34, 37, 38]. None of the RCTs reported blinding of participants. Almost all the five non-randomised studies showed a low risk of bias for selection of participants, measurement of interventions, departures from intended interventions. However, how confounding factors were controlled for and missing data were dealt with was not reported in any of the studies [31, 32, 35, 36, 39]. The measurement of outcomes in all the studies was based on a self-reported measure except for one study

[37]. One of the non-randomised studies had significant technological challenges and inadequacies concerning the design of text message content, which could have influenced the outcome of the study [35]. Finally, only five studies out of ten analysed by intention-to-treat i.e. analysed the participants according to the groups to which they were originally assigned [29, 33, 34, 37, 38], which provides a more accurate, unbiased estimate of the findings reported in this study.

**mHealth interventions and platforms used.** The most used mHealth platform was the Short Message Service (SMS) (9 studies). Only one of the studies used an interactive web-based peer support platform [31]. Five studies were based on unidirectional and interactive messaging services; three were based solely on 2-way interactive and two on unidirectional messaging services. The interventions focused on shaping knowledge and increasing the use of reproductive health interventions or services. Two studies evaluated SRH knowledge [29, 35], four assessed contraceptive use/birth control [29, 33, 34, 39], three examined pregnancy and fertility intentions [29, 33, 34]. One focused on facility childbirth delivery [33], two on exclusive breastfeeding (EBF) [33, 34], four on HIV Antiretroviral Therapy (ART) adherence [31, 32, 37, 38], and two on sexual behaviour [29, 34].

## Effectiveness of the mHealth interventions on improving SRH outcomes

**SRH knowledge.** Table 2 shows the effect of the intervention on SRH knowledge, sexual behaviour, and contraceptive use. Three studies examined adolescent SRH knowledge outcomes [29, 31, 35]. Only one study evaluating a unidirectional and interactive intervention among adolescent girls showed a positive effect at 3-months and 15 months after controlling for covariates, (age, religion, ethnicity, parents' educational level [29]. Sexual and Reproductive Health (SRH) knowledge increased by 11% (95% Confidence Interval (CI): 7%, 15%) and 24% (95% CI: 19, 28), greater than in the control group, respectively. This effect was maintained at 15 months in the interactive group, however, ceased to be significant among the unidirectional group. The remaining two studies showed a difference in knowledge between the intervention and control groups [31, 35]. However, the intervention group in one of the studies reported more false answers on SRH knowledge about STIs, abortion & contraception (1.7) compared to the control group (1.9) [35]. Similarly, with reference to improvement in HIV/ART knowledge, there was no statistically significant change in knowledge among adolescents who participated in the interactive web-based intervention (adolescents demonstrated less knowledge at end-line comparing to baseline) and those that did not take part [31].

**Sexual health behaviour.** Two studies reported effects on adolescent's sexual behaviour [29, 34]. One study found that the SMS intervention influenced young women's resumption of sexual intercourse such that most women (31.8% at 6 weeks, 57.9% at 14 weeks, and 67.7% at 6 months) reported having resumed sexual intercourse by 6 months [34]. Similarly, in another study, the interactive intervention was positively associated with having sex without a condom among sexually active adolescents in the interactive group (OR = 3.47; 95% CI = 1.12, 10.74). However, the intervention did not influence the age of sexual debut for those who have ever had sexual intercourse [29]. (See Table 2).

**Contraceptive/Birth control access and use.** Four studies reported on contraceptive use [29, 33, 34, 39]. Evidence from the four studies showed that the intervention increased the use and access to contraceptive services and family planning initiation among adolescents and this was higher among the interactive compared with unidirectional group (Table 2). One study found that highly effective contraceptive (HEC) use at 6 months postpartum was significantly higher among those in the SMS group (69.9%) than in the control group (57.4%) ([RR] = 1.22; 95%; 1.01, 1.47; P = .04) [32]. Another study reported that contraceptive use was significantly

higher in both intervention arms by 16 weeks (1-way SMS: 72% and 2-way SMS: 73%; p = 0·03 and 0·02 versus 57% control, respectively) [33]. This trend was reported in another study which found that the interactive intervention increased the odds of using oral contraceptives (OR = 13.2; 95% CI = 1.08, 161) and decreased the odds of using emergency contraception (OR = 0.22; 95% CI = 0.05, 0.88) [29]. Likewise, in another study, participants who engaged with the intervention accessed contraceptive information more frequently than non-intervention group [39].

**Antiretroviral therapy (ART) adherence.** Table 2 presents the results of the effects of the interventions on antiretroviral therapy adherence (ART), pregnancy & childbirth and breast feeding. Four studies reported effects on ART adherence [31, 32, 37, 38]. Overall, the four studies showed an improvement in adherence and pre-exposure prophylaxis (PrEP) initiation. However, this improvement was not statistically significant except for one study [32]. Despite relying on a routine collected measure, an evaluative study found that women who enrolled in the intervention were almost twice more likely to continue PrEP (22% vs. 43%; aRR = 1.75; 95% CI = 1.21, 2.55; P = .003), than women who initiated PrEP in the month before the intervention implementation [32]. This is contradicted by another study which showed no statistical difference in adherence between the intervention and control groups (Adherence was 64% for the 1-way group [OR = 0.64; 95% CI: 0.58, 0.70; P = .27] and 61% for the 2-way group [0.56, 0.67; P = .15], compared with 67% in the control group (OR = 0.67; 95% CI:0.62, 0.72] [35]. Also, there was no statistically difference between the proportion of participants achieving adherence of at least 90% over the 48-week period of analysis (1-way group = 28%; 2-way group = 26% and control = 29%; P = .85 and .69, respectively). A similar study found that at baseline, 71.6% of participants reported not to have missed any doses in the last week, while 77.8% of the participants at the post intervention reported not to have missed any doses in the last week [31]. Although not statistically significant (p = 0.95) and finally, this level of insignificance persists in another study, which found that after controlling for baseline adherence, the intervention group 1 (T1) had 3.8% lower adherence than the control group (95% CI -9.9, 2.3) and the Intervention group 2 (T2) had 2.4 percentage points higher adherence than the control group (95% CI -3.0, 7.9). However, the differences were not statistically significant for either intervention groups [38].

**Pregnancy and childbirth.** Three studies reported the effect of the intervention on pregnancy and childbirth outcomes [29, 33, 34]. These RCTs studies showed that the intervention influenced fertility intentions, reduced the odds of self-reported pregnancy and facility delivery. However, the effects were not statistically significant between the intervention and the control groups except for one study [29], where both the unidirectional and the interactive groups significantly lowered the odds of self-reported pregnancy by 86% in the adjusted models (odds ratio [OR] = 0.14; 95% CI = 0.03, 0.71) and 85% (OR = 0.15; 95% CI = 0.03, 0.86), respectively, compared with the control group. In another study of 184 participants who initially wanted to become pregnant again and whose fertility intentions were similar, found that after 6-month visits, fertility intentions were similar between groups with 26.2% who reported a desire to stop childbearing [34]. A similar study stated that at 10 weeks, facility delivery was high in all 3 intervention arms [33]. Among 277 women providing delivery data, 273 (98.6%) reported delivering in a facility, with no difference between the 1-way and control arms [relative risk (RR) 1.00, 95% CI 0·97–1·03; p = 0·99] or 2-way and control arms [RR 0·99, 95% CI 0·95–1·03; p = 0·54]. Although there were apparently fewer still-births and infant deaths in the 2-way group compared to the control group (3·1% versus 8%), but not statistically significant (p = 0.21) [31]. No serious adverse events occurred because of the intervention although one maternal death occurred (See Table 3).

**Breastfeeding.** As reported in Table 3, only two RCTs reported the effects of SMS intervention on exclusive breastfeeding (EBF) [33, 34]. Both studies reported inconsistent findings with one study showing a significant improvement in EBF among the intervention group than the control group [33], and another showing no significant difference across the two groups examined [34]. One of the studies revealed that women in both intervention arms were significantly more likely to report EBF at 10 and 16 weeks than women in the control arm [10 weeks: Control arm (RR: 0.79; CL:0.69-.86); 1-way SMS RR = 0.93 (CI:0.86–0.97; p = 0.003), 2-way SMS: RR = 0.96 (0.89–0.98; P = 0.0004); At 16 weeks: Control arm [RR = 0.62; CI: 0.52–0.71]; 1-way SMS [RR = 0.82; CI 0.72–0.89, P = 0.002], 2-way SMS [RR = 0.93, CI: 0.85–0.97; P = 0.001]. At 24 weeks, the probability of EBF was higher in both intervention groups than in the control, but only statistically significant in the 2-way messaging group [0·49 in 1-way, 0·62 in 2-way, and 0·41 in control, (p = 0·30 and 0·005 for 1-way and 2-way vs. control, respectively).

## Components and characteristics of mHealth interventions

**Behavioural change components of the interventions.** Overall, 23 from a possible list of 93 BCTs were identified as intervention components in the included studies (Fig 2). The 23 BCTs were from six out of the 16 possible domains (feedback & monitoring, social support, shaping knowledge, natural consequences reward & threat) of BCTs [23]. The most commonly used BCTs in these studies were feedback & monitoring, and social support (6 studies). The feedback and monitoring techniques mostly used in the studies focused on monitoring and providing informative feedback on scores and performance of the behaviour among participants. However, the feedback was based on change in knowledge and not on change in behaviour. Half of the studies (five out of ten) described how participants were socially and practically supported to achieve the intervention objectives, although some of the studies did not specify the nature of social support provided, and four studies reported on shaping knowledge through instruction on how to perform a behaviour. Two interventions did not report the use of any BCTs [36, 39].

## Behavioural Change Techniques (BCTs) domains

| Author's names & year | 1 | 2 | 3 | 4 | 5 | 6 | 7 | 8 | 9 | 10 | 11 | 12 | 13 | 14 | 15 | 16 |
|---|---|---|---|---|---|---|---|---|---|---|---|---|---|---|---|---|
| Cele and Archary 2019 | 0 | 0 | 0 | 0 | 0 | 0 | 0 | 0 | 0 | 0 | 0 | 0 | 0 | 0 | 0 | 0 |
| De-Kruijf et al 2016 | 0 | 0 | 0 | 1 | 0 | 0 | 0 | 0 | 0 | 0 | 0 | 0 | 0 | 0 | 0 | 0 |
| Harrington et al., 2019 | 0 | 1 | 1 | 1 | 1 | 0 | 1 | 0 | 0 | 0 | 0 | 0 | 0 | 0 | 0 | 0 |
| Ivanova 2019 | 0 | 1 | 0 | 0 | 0 | 0 | 0 | 0 | 0 | 0 | 0 | 0 | 0 | 0 | 0 | 0 |
| L'Englea et al 2012 | 0 | 0 | 0 | 0 | 0 | 0 | 0 | 0 | 0 | 0 | 0 | 0 | 0 | 0 | 0 | 0 |
| Linnemayr et al., 2017 | 0 | 1 | 1 | 0 | 0 | 0 | 0 | 0 | 0 | 0 | 0 | 0 | 0 | 0 | 0 | 0 |
| MacCarthy et al., 2020 | 0 | 1 | 0 | 0 | 0 | 0 | 0 | 0 | 0 | 0 | 0 | 0 | 0 | 0 | 0 | 0 |
| Pintye et al., 2020 | 0 | 1 | 1 | 0 | 1 | 0 | 0 | 0 | 0 | 0 | 0 | 0 | 0 | 0 | 0 | 0 |
| Rokicki et al., 2016; | 0 | 1 | 1 | 1 | 0 | 0 | 1 | 0 | 0 | 1 | 0 | 0 | 0 | 0 | 0 | 0 |
| Unger et al 2018 | 0 | 0 | 1 | 1 | 0 | 0 | 0 | 0 | 0 | 0 | 0 | 0 | 0 | 0 | 0 | 0 |

**Fig 2. Heat map: Showing the behavioural techniques used as intervention components in each study.** Key: 1 = Goals & planning, 2 = Feedback & monitoring, 3 = Social support, 4 = Shaping knowledge, 5 = Natural consequences, 6 = Comparison of behaviour, 7 = Associations 8 = Repetition & substitution, 9 = Comparison of outcomes, 10 = Reward & threat, 11 = Regulation, 12 = Antecedents, 13 = Identity, 14 = Scheduled consequences. 15 = Self-belief, 16 = Covert learning.

The second aspect was to identify the mHealth intervention content for the included studies: where it is being implemented (context), and how it was implemented (technical features) to support replication of the intervention using the mHealth evidence reporting and assessment (mERA) guidelines [24]. Fig 3 below shows the number of included studies meeting each mHealth criterion. On average, about 35% (6%-63%) of the 16 mERA criteria was achieved among all the 10 studies. Overall, most studies described the mode and frequency of intervention delivery [29, 31–39], how people were informed of the programme [29, 31–39], how the content of the intervention was developed [29, 31–34, 36–39] and the technology platform/ software used in the programme implementation [29, 31–34, 37–39]. However, there were limited information on the barriers/challenges faced by participants in adopting the intervention [36] study), the physical infrastructure used to support the interventions [32, 34] and the security and confidentiality protocol of the interventions [33, 34].

## Secondary outcomes

**Acceptability of mHealth interventions.**   Four studies evaluated adolescent acceptability of receiving an mHealth intervention for increasing adherence to HIV prevention or treatment [31, 32, 36, 38]. Three studies were among HIV positive adolescents [31, 36, 38] and one was pre-exposure prophylaxis among pregnant or post-partum women [32]. No study reported on the acceptability of mHealth interventions by parents of adolescents.

Across all four of the studies, participants showed a positive attitude, and were willing to use and recommend mHealth interventions to others. One study reported that almost all the women (95%) would recommend the intervention (mWACh-PrEP) to other women who use pre-exposure prophylaxis (PrEP), and 95% would also use the program again if offered [32].

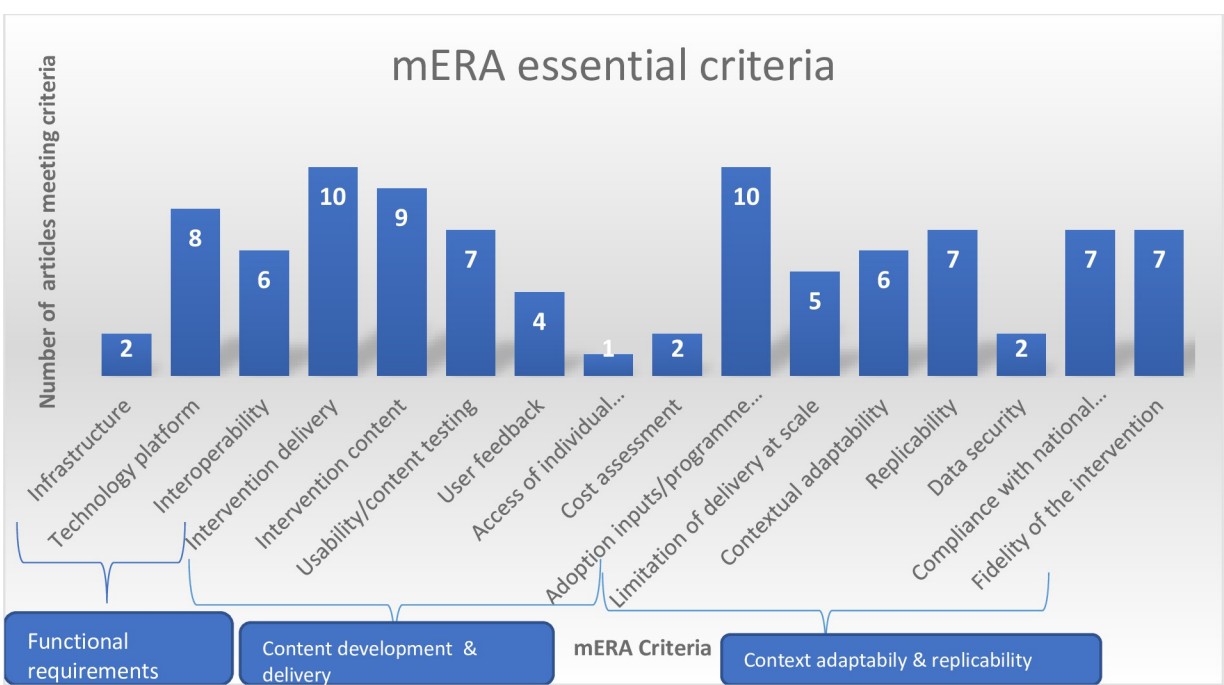

**Fig 3. Number of papers that met mERA essential criteria among the 10 selected studies.** No of studies in each component: *Infrastructure* [32, 42]; *Technology platform* [29, 31–34, 37–39]; *Interoperability* [29, 31–34, 38]; *Intervention delivery* [29, 31–39]; *Intervention content* [29, 31–34, 36–39]; *Usability testing* [29, 31–35, 37]; *User feedback* [31, 32, 38, 39]; *Access of individual participants* [38]; *Cost assessment* [29, 34]; *Adoption inputs/programme entry* [29, 31–39]; *Limitations for delivery at scale* [29, 35–38]; *Contextual adaptability* [31–35, 38]; *Replicability* [29, 32–34, 37, 38]; *Data security* [33, 34]; *Compliance* [29, 32, 34, 36, 38, 39]; *Fidelity of the intervention* [29, 32–34, 37–39].

Also, 94% and 89% of women reported that SMS helped them understand and adhere to PrEP, respectively [32].

Another study found that among 81 adolescents with HIV that completed a mid-term questionnaire, the majority (95%) agreed to use the web-based interventions (ELIMIKA website) again and 87% would recommend it to others [31]. A related study in Uganda, showed that participants had positive attitudes about SMS as an incentive to adherence to ART (SITA) [38]. At follow-up, 96.6% of adolescents with HIV reported that they would remain in the intervention group if they had the choice (95.3% in the treatment 1 (T1) group and 97.8% in treatment 2 (T2) group, and 84.2% said there was nothing about SITA that they did not like (86.0% in T1 and 82.6% in T2). Participants from both intervention groups felt that SITA boosted their morale and prompted them to take their ART medication on time. Despite a small sample size, evidence from another study revealed that 65% of adolescents with HIV were willing to participate in a mHealth intervention to support treatment adherence [36].

**Feasibility of delivery of mHealth interventions.** Three studies reported on the feasibility of delivering mHealth intervention to adolescents in SSA [36, 38, 39]. All three of the studies established feasibility of delivering HIV treatment adherence and contraceptives information to adolescents using mHealth interventions. One study showed that sending text messages with information on a participant's own adherence, information about the adherence performance of their peers and the recruitment process was practicable among HIV positive youth [38]. Also, a study found that use of text messages to support treatment adherence in adolescents with HIV was feasible, especially among in-school adolescents with high ownership of mobile phone with 67% willing to receive health related SMS [36]. Another study demonstrated that text messages comprising comprehensive information on contraceptive methods can be feasibly delivered and accessed by men and women of reproductive age [39]. However, Fig 3 (mERA checklist) above shows that only five studies reported the appropriateness of the intervention to the context and any possible adaptations required.

**Cost-effectiveness of mHealth interventions.** None of the included studies reported cost-effectiveness outcomes. Intervention costs was among the least reported components in the mERA checklist (Fig 3). One study reported that the marginal costs of the interactive and unidirectional component per participant were US $1.91 and US $0.30, respectively [29]. Another study reported that the intervention was "relatively inexpensive "but with no information was provided on the specific costs of the intervention [34].

## Discussion

Overall, the review demonstrates that mHealth interventions improve adolescent's uptake of SRH services across a wide range of services. The evidence was strongest for increasing adolescent's use of contraceptives. This is consistent with the results of previous reviews outside SSA [18, 19, 40]. For other SRH outcomes, the evidence was inconsistent. There was only one study that demonstrated a significant effect of mHealth interventions for each of the following outcomes: improving sexual health knowledge, adherence to HIV treatment, self-reported pregnancy, exclusive breastfeeding, delay of resumption of sexual activities for postpartum young women and increase in health facility delivery among adolescents, which is insufficient to establish the effectiveness of the interventions on these outcomes. Evidence from previous reviews conducted in high- and middle-income countries shows that mHealth interventions significantly improve SRH knowledge among adolescents [19]. Surprisingly, while there was an improvement on adolescent's uptake of SRH services across a wide range of studies, one of the studies indicated increase in sex without having a condom among sexually active adolescents in the intervention group. The reason for this is quite unclear, and could be an artefact

given that it was only one out of ten studies that reported negative effects of mHealth in this review.

Most of the studies that had significant effects on improving uptake of SRH services among adolescents were those with two-way interactive components rather than one-way messaging services. Furthermore, interventions with more BCTs showed stronger efficacy than those with limited BCTs. This indicates that integration of effective BCTS and interactive components in future mHealth interventions may lead to more effective interventions [41]. The non-significant effects of some of the interventions in improving uptake of SRH services by adolescents in this study could arguably be attributable to the limited active ingredients of BCTs in these studies.

Our results show that only 23 out of possible 93 BCTs were captured in the included studies and in some cases, there was no single element of BCTs in the intervention. The integration of active BCTs ingredients plays an important part in ensuring the interventions exert their effect [42] and bring about the desired change in the target behaviour [43]. This is because previous studies have shown that BCTs have been identified for interventions that prevent sexually transmitted infections (STIs) [44] and improve use of condom [45]. Our results showed that only few studies reported the challenges faced by participants in adopting the interventions, the physical infrastructure used to support the interventions and the security and confidentiality protocol of the interventions. This is concerning as information on these features could aid effective design of future mHealth intervention. The lower level of reporting completeness on essential features of mHealth interventions has been reported in previous reviews [19]. Although, the limited reporting could be attributable to the fact that the WHO developed mHealth reporting guideline is fairly new and insufficient reporting of mHealth features in studies published before the guideline was developed may be expected [19].

Finally, the results of our review showed that mHealth interventions to promote treatment adherence to prevent or treat HIV were acceptable to individuals, and can feasibly be delivered among adolescents in SSA. However, four of the five studies were non-randomised with moderate risk of bias.

Despite the potential for digital interventions to be scalable and delivered at low cost, cost-effectiveness was not evaluated in any of the included studies. Furthermore, the cost implications of these mHealth interventions were among the least reported components of the mERA checklist. For the costs that were reported it was unclear if they referred to development costs, delivery costs or a combination of both. A similar review on the effectiveness of digital interventions on improving physical activity among adolescents also showed that none of the 32 included studies reported the cost effectiveness of the interventions [46].

## Strengths and limitations

Overall, the review followed an established guideline for undertaking reviews [25]. The literature search was comprehensive and identified a high number of potential studies including search of grey literature sources. The screening process was carried out by three independent reviewers, minimizing the risk of missing relevant studies. Data extraction and quality assessment were rigorous and transparent. Our findings were largely informed by high quality RCTs and non-randomised studies with low risk of bias. However, the measurement of outcomes in most of the studies was based on a self-reported measure, which could have introduced bias, which may have overestimated the treatment effect. It is important to note that few studies included older women in their analysis. For example, one of the papers [34] included women aged 14 and above in their intervention; making it difficult to disaggregate the data for young women aged 14–24 years. Also, caution should be exercised when interpreting the findings of

the non-randomised studies given that most of them did not account for the missing data or controlling for confounding.

## Conclusions

This review demonstrates that interactive mobile health interventions with effective behaviour change techniques have strong potential to improve adolescent uptake of health services. This evidence heightens the need to develop mHealth interventions tailored for adolescents, which are theoretically informed and incorporate effective behaviour change techniques. Such interventions could improve the use of sexual and reproductive health services and lead to health improvement among adolescents in SSA. Also, future research should prioritise transparent reporting of the essential components of mHealth interventions to support accurate generalisation, application of the findings, and replication of the intervention. Studies evaluating cost-effectiveness of mhealth interventions are required.

## Supporting information

**S1 Checklist.**
(DOCX)

**S1 File.**
(DOCX)

## Author Contributions

**Conceptualization:** Lesley Smith, Dan Kaseje, Monica Magadi.

**Data curation:** Franklin I. Onukwugha, Lesley Smith, Charles Wafula, Margaret Kaseje, Mark Hayter.

**Formal analysis:** Franklin I. Onukwugha, Lesley Smith.

**Funding acquisition:** Monica Magadi.

**Investigation:** Franklin I. Onukwugha, Lesley Smith, Dan Kaseje, Charles Wafula, Margaret Kaseje, Monica Magadi.

**Methodology:** Franklin I. Onukwugha, Lesley Smith, Dan Kaseje, Margaret Kaseje, Bev Orton, Mark Hayter.

**Supervision:** Monica Magadi.

**Writing – original draft:** Franklin I. Onukwugha.

**Writing – review & editing:** Lesley Smith, Dan Kaseje, Charles Wafula, Margaret Kaseje, Bev Orton, Mark Hayter, Monica Magadi.

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
