## [Decision Letter · Decision Letter 0]

26 Oct 2021

PONE-D-21-19913The effectiveness and characteristics of mHealth interventions to increase adolescent’s use of Sexual and Reproductive Health services in Sub-Saharan Africa: A systematic reviewPLOS ONE

Dear Dr. Onukwugha,

Thank you for submitting your manuscript to PLOS ONE. After careful consideration, we feel that it has merit but does not fully meet PLOS ONE’s publication criteria as it currently stands. Therefore, we invite you to submit a revised version of the manuscript that addresses the points raised during the review process. The article is a systematic review on a previously understudied topic. The authors include the mandatory PRISMA checklist and diagram. The article was evaluated by 2 expert reviewers. The AMSTAR guidelines, see http://amstar.ca/index.php and http://amstar.ca/docs/AMSTARguideline.pdf., are also used to help assess the quality of systematic reviews.

In this particular instance, in addition to other minor edits/suggestions, the reviewers note:

That the inclusion and exclusion criteria are not clear enough.That the evidence on which the bias assessment is carried out are not explicit or systematic enough in the text, and that in some cases the limitations of the included studies are not explicit enough.  That the discussion and conclusion section may not be supported by the data. There is a discrepancy between the mixed results reported and the contention that the interventions were cost-effective and improved use.

We look forward to receiving your revised manuscript.

Kind regards,

José Antonio Ortega, Ph.D.

Academic Editor

PLOS ONE

Reviewers' comments:

Reviewer's Responses to Questions

**Comments to the Author**

1. Is the manuscript technically sound, and do the data support the conclusions?

Reviewer #1: Partly

Reviewer #2: Partly

2. Has the statistical analysis been performed appropriately and rigorously? 

Reviewer #1: N/A

Reviewer #2: N/A

3. Have the authors made all data underlying the findings in their manuscript fully available?

Reviewer #1: No

Reviewer #2: Yes

4. Is the manuscript presented in an intelligible fashion and written in standard English?

Reviewer #1: Yes

Reviewer #2: Yes

5. Review Comments to the Author

Reviewer #1: Overall summary

This review examines the evidence for mHealth interventions to increase adolescent’s use of sexual and reproductive health services in sub-Saharan Africa. The authors conclude that mHealth interventions are effective and cost-effective; however, my interpretation of the results is that they were mixed, and it is not so clear that they were effective or cost-effective overall. More information regarding the risk of bias assessment is needed to fully interpret the results.

Abstract

Clear summary of research.

As mentioned above, the authors conclude that mHealth interventions are cost-effective and effective among adolescents in SSA; however, my interpretation of the results is that they were mixed and it is not so clear that they were effective overall.

Minor grammatical errors need correcting.

Introduction

Good summary of the evidence and justification for undertaking the systematic review.

Clear definition of mHealth. A definition of sexual and reproductive health would also be helpful here.

The authors state that sub-Saharan Africa “had the highest prevalence of adolescent pregnancy in the world” (line 48) but do not state which year or when they are referring to.

It would be helpful to know the coverage of mobile phone subscribers (line 51 mentions 600 million mobile phone subscribers, but do we know how many will not have access to mobile phones?).

Although the aim seems clear, it is a little unclear whether the authors were also intending to review the evidence for improving sexual and reproductive health in addition to service use (objective ii) as these are not necessarily the same thing.

Methods

A comprehensive search was undertaken of a number of different databases.

A clear definition of the population would be helpful here (for example, the age range considered to be adolescent), possibly in a PICO format. This could also expand on the inclusion criteria (for example, study design and outcomes).

No indication of restrictions by publication year or language – I note this is mentioned in the protocol, but it would be helpful to know if this was followed and whether any non-English language papers were identified, and if so, who translated those manuscripts.

Lines 146-147 state that “statistical pooling is not possible in this study” but a few words explaining why would be helpful for readers.

The supplementary search strategy describes the MEDLINE search strategy that is provided as the “preliminary search strategy” – was this the final search strategy used for MEDLINE? How were the three concepts combined?

Results

Table 1 describes the sample description. There does not seem to be a definition of adolescents, but I note the age ranges of the included papers vary widely and seem to include older adults too (e.g. Harrington et al is described as 14 years and above so does that include all adults?). Clarification is needed of the population included. As a minor point, there is a need to ensure consistency in formatting of Table 1.

It is good to see a PRISMA flow diagram, but it is not clear from this where the 132 records were identified from that were not from database searching, and whether the 10 included studies were identified from database searching or by other means which would be helpful to know.

There is limited information about risk of bias. Without this, it is difficult to interpret the results. A table or similar with the full risk of bias assessment for each paper would be helpful in interpreting the strength of the evidence. Similarly, I could not identify information regarding uptake of the intervention in control/intervention groups and whether intention to treat analyses were undertaken, which would again affect interpretation.

Figure 3 is helpful to show how the studies adopted the mERA essential criteria. It could be helpful to show Figure 3 by individual studies, so that the reader is able to see how well individual studies conformed to mERA criteria, given the different outcomes included in each study and the heterogeneity of studies overall.

For the pregnancy and childbirth section, I note odds of self-reported pregnancy are given (line 270) but it is unclear to me whether this is unintended pregnancy or intended pregnancy or intention unknown?

For the cost-effectiveness section (line 358), it would be helpful to describe how strong the evidence was for cost-effectiveness. A marginal cost is reported, but again it would be helpful to see what outcome this was referring to. Given that most studies did not report cost-effectiveness, it seems that this systematic review cannot interpret much in terms of the cost effectiveness of mHealth interventions in SRH overall.

Discussion

Well written discussion overall.

Potential harms of mHealth are not discussed in any detail, yet one study showed increased sex without a condom in the intervention arm so this could be a risk.

The fact that non-randomised studies did not report controlling for confounding or accounting for missing data (lines 169-170) seems to me to be a significant issue and perhaps should be mentioned in the discussion in more detail as this will have substantial implications for interpretation of the results of those studies.

In terms of the mERA checklist, the authors note that completeness of reporting is poor. It seems concerning that only 2 included information on infrastructure, and only 1 included access of individual participants in terms of barriers and facilitators to the adoption of the intervention among study participants. This seems a considerable limitation of included studies and it is surprising not to see this discussed in more detail in the discussion.

Reviewer #2: The authors conducted a systematic review of the literature to assess the effectiveness of mobile health (mHealth) interventions in facilitating the uptake of sexual and reproductive health services (SRH) among adolescents in sub-Saharan Africa (SSA). They reported on ten studies that met their review criteria. They concluded that mHealth interventions were effective in improving adolescents’ use of SRH services.

Generally, the paper is well written and contributes to knowledge on the use of mHealth for SRH services. The study is particularly relevant because it focuses on adolescents, a population mostly challenged with sexual and reproductive health issues in SSA.

However, there are few areas for improvement.

First, the inclusion and exclusion criteria used in this study are not entirely clear. I will suggest the authors explicitly state this in the paper.

Secondly, the discussion section could be generally improved. The discussion is an opportunity to provide explanations of the results and the possible implications. There is a need to provide a better explanation of how some of the conclusions stated in this section were reached. For example, four out of 10 studies reported acceptability, three reported feasibility, and two reported cost-effectiveness. However, you stated in line 396 that “Finally, the results of our review showed that mHealth interventions were cost-effective, acceptable and can feasibly be delivered among adolescents in SSA.” Are the findings strong enough to support this assertion? If so, please explain.

Minor issues:

I suggest the heading “mHealth platforms used” be re-titled as “mHealth interventions and platforms used.” Still, under this section heading, you mentioned eight studies made use of SMS. However, it is not clear what mHealth platforms the other two studies used. It might be good to account for all the reviewed studies in the explanation.

The first secondary objective states, “Acceptability of mHealth interventions to adolescents, and parents in providing SRH information in SSA.” However, when reporting the findings for this objective, there was no mention of parents. It might be good to consider this when discussing your findings here.

Line 43 – “…young people face enormous barriers accessing Sexual and Reproductive Health (SRH) information and services.” Would you please highlight some of these barriers?

6. PLOS authors have the option to publish the peer review history of their article (what does this mean?). If published, this will include your full peer review and any attached files.

Reviewer #1: No

Reviewer #2: No

---

## [Author Response · Author response to Decision Letter 0]

10 Dec 2021

Thank you for your kind and thorough review of our manuscript. We have carefully revised the manuscript in line with your suggestions and have addressed all the concerns raised. The response to reviewers' document and the revised manuscripts including the mERA checklist (figure 3) have been uploaded as well.

---

## [Editor Report · Decision Letter 1]

15 Dec 2021

The effectiveness and characteristics of mHealth interventions to increase adolescent’s use of Sexual and Reproductive Health services in Sub-Saharan Africa: A systematic review

PONE-D-21-19913R1

Dear Dr. Onukwugha,

We’re pleased to inform you that your manuscript has been judged scientifically suitable for publication and will be formally accepted for publication once it meets all outstanding technical requirements.

Kind regards,

José Antonio Ortega, Ph.D.

Academic Editor

PLOS ONE

Additional Editor Comments (optional):

The changes from the previous version address satisfactorily, in the opinion of the editor, the shortcomings of the previous version and the work is acceptable for publication without the need of sending itt back to the reviewers. My congratulation to the authors.

One minor thing to change. The heading of figure 3 is incorrect and it needs to be edited. It reads "Figure 3: Percentage of papers that met mERA essential criteria". But it does not show any percentage. It should be rewritten as "Number of  papers that met mERA essential criteria among the 10 selected studies" or something along these lines.
---

## [Editor Report · Acceptance letter]

19 Dec 2021

PONE-D-21-19913R1 

The effectiveness and characteristics of mHealth interventions to increase adolescent’s use of Sexual and Reproductive Health services in Sub-Saharan Africa: A systematic review 

Dear Dr. Onukwugha:

I'm pleased to inform you that your manuscript has been deemed suitable for publication in PLOS ONE. Congratulations! Your manuscript is now with our production department. 

Kind regards, 

on behalf of

Dr. José Antonio Ortega 

Academic Editor

PLOS ONE